# 3D-IntPhys: Learning 3D Visual Intuitive Physics for Fluids, Rigid Bodies, and Granular Materials

## Abstract

Given a visual scene, humans have strong intuitions about how a scene can evolve over time under given actions. The intuition, often termed visual intuitive physics, is a critical ability that allows us to make effective plans to manipulate the scene to achieve desired outcomes without relying on extensive trial and error. In this paper, we present a framework capable of learning 3D-grounded visual intuitive physics models purely from unlabeled images. Our method is composed of a conditional Neural Radiance Field (NeRF)-style visual frontend and a 3D point-based dynamics prediction backend, in which we impose strong relational and structural inductive bias to capture the structure of the underlying environment. Unlike existing intuitive point-based dynamics works that rely on the supervision of dense point trajectory from simulators, we relax the requirements and only assume access to multi-view RGB images and (imperfect) instance masks. This enables the proposed model to handle scenarios where accurate point estimation and tracking are hard or impossible. We evaluate the models on three challenging scenarios involving fluid, granular materials, and rigid objects, where standard detection and tracking methods are not applicable. We show our model can make long-horizon future predictions by learning from raw images and significantly outperforms models that do not employ an explicit 3D representation space. We also show that, once trained, our model can achieve strong generalization in complex scenarios under extrapolate settings.

## 1 Introduction

Humans can achieve a strong intuitive understanding of the 3D physical world around us simply from visual perception (Baillargeon et al., 1985; Battaglia et al., 2013; Spelke, 1990; Smith et al., 2019; Sanborn et al., 2013; Carey & Xu, 2001). As we constantly make physical interactions with the environment, the intuitive physical understanding applies to objects of a wide variety of materials (Bates et al., 2018; Ullman et al., 2019). For example, after watching videos of water pouring and doing the task ourselves, we can develop a mental model of the interaction process and predict how the water will move when we apply actions like tilting or shaking the cup (Figure 1). The ability to predict the future evolution of the physical environment is extremely useful for humans to plan our behavior and perform everyday manipulation tasks. It is thus desirable to develop computational tools that learn 3D-grounded models of the world purely from visual observations that can generalize to objects with complicated physical properties like fluid and granular materials.

There has been a series of works on learning intuitive physics models of the environment from data. However, most existing work either focuses on 2D environments (Watter et al., 2015; Agrawal et al., 2016; Fragkiadaki et al., 2016; Xu et al., 2019; Finn & Levine, 2017; Babaeizadeh et al., 2021; Qi et al., 2021; Kipf et al., 2019; Ye et al., 2019b; Hafner et al., 2019b;a; Schrittwieser et al., 2020; Li et al., 2016; Finn et al., 2016; Lerer et al., 2016; Veerapaneni et al., 2019; Girdhar et al., 2020; Chang et al., 2016; Xue et al., 2016) or has to make strong assumptions about the accessible information of the underlying environment (Li et al., 2019a; 2020; Sanchez-Gonzalez et al., 2020; Pfaff et al., 2021; Zhang et al., 2016; Tacchetti et al., 2018; Sanchez-Gonzalez et al., 2018; Battaglia et al., 2018; Ajay et al., 2019; Janner et al., 2019a) (e.g., full-state information of the fluids represented as points). The limitations prevent their use in tasks requiring an explicit 3D understanding of the environments and

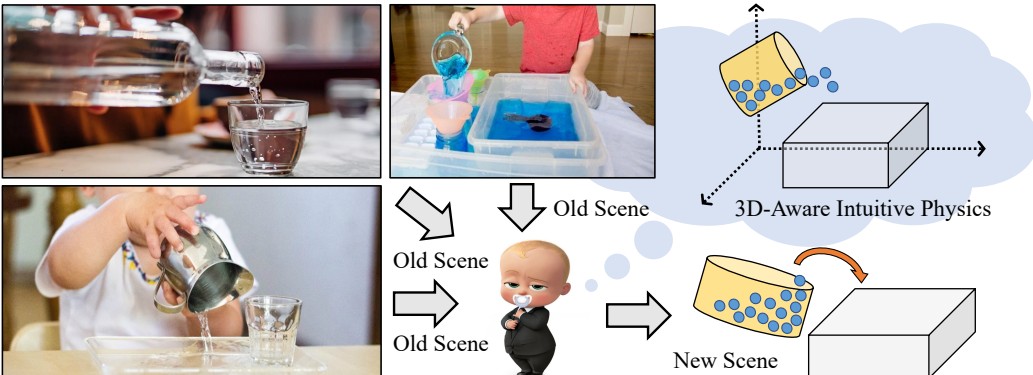

Figure 1: **Visual Intuitive Physics Grounded in 3D Space.** Humans have a strong intuitive understanding of the physical environment. We can predict how the environment would evolve when applying specific actions. This ability roots in our understanding of 3D and applies to objects of diverse materials, which is essential when planning our behavior to achieve specific goals. In this work, we leverage a combination of implicit neural representation and particle representation to build 3D-grounded visual intuitive physics models of the world that applies to objects with complicated physical properties, such as fluids, rigid objects, and granular materials.

make it hard to extend to more complicated real-world environments where only visual observations are available. There are works aiming to address this issue by learning 3D-grounded representation of the environment and modeling the dynamics in a latent vector space (Li et al., 2021b;a). However, these models typically encode the entire scene into one single vector. Such design does not capture the structure of the underlying systems, limiting its generalization to compositional systems or systems of different sizes (e.g., unseen container shapes or different numbers of floating ice cubes).

In this work, we propose 3D Visual Intuitive Physics (3D-IntPhys), a framework that learns intuitive physics models of the environment with explicit 3D and compositional structures, purely from visual observations. Specifically, the model consists of (1) a perception module based on conditional Neural Radiance Fields (NeRF) (Mildenhall et al., 2020; Yu et al., 2021) that transforms the input images and instance masks into 3D point representations and (2) a dynamics module instantiated as graph neural networks to model the interactions between the points and predict their evolutions over time. Despite advances in graph-based dynamics networks (Sanchez-Gonzalez et al., 2020; Li et al., 2019a), existing methods require strong supervision provided by 3D GT point trajectories, which are hard to obtain in most real setups. To tackle the problem, we train the dynamics model using (1) a distribution-based loss function measuring the difference between the predicted point sets and the actual point distributions at the future timesteps and (2) a spacing loss to avoid degenerated point set predictions. Our perception module learns spatial-equivariant representations of the environment grounded in the 3D space, which then transforms into points as a flexible representation to describe the system's state. Our dynamics module regards the point set as a graph and exploits the compositional structure of the point systems. The structures allow the model to capture the compositionality of the underlying environment, handle systems involving objects with complicated physical properties (e.g., fluid and granular materials), and perform extrapolated generalization, which we show via experiments greatly outperform various baselines without a structured 3D representation space.

## 2 RELATED WORK

**Visual dynamics learning.** Existing works learn to predict object motions from pixels using frame-centric features (Agrawal et al., 2016; Finn & Levine, 2017; Babaeizadeh et al., 2021; Hafner et al., 2019b;a; Suh & Tedrake, 2020; Lee et al., 2018; Vondrick et al., 2015; Zhang et al., 2018; Burda et al., 2019; Hafner et al., 2019c; Wu et al., 2021) or object-centric features (Fragkiadaki et al., 2016; Watters et al., 2017; Kipf et al., 2019; Qi et al., 2021; Janner et al., 2019b; Veerapaneni et al., 2019; Ding et al., 2020; Girdhar et al., 2020; Riochet et al., 2020; Ye et al., 2019a), yet, most works only demonstrate the learning in 2D scenes with objects moving only on a 2D plane. We argue that one reason that makes it hard for these existing methods to be applied to general 3D visual scenes is because they often operate on view-dependent features that can change dramatically due to changes in the camera viewpoint, which shouldn't have any effect on the actual motion of the objects. Recent

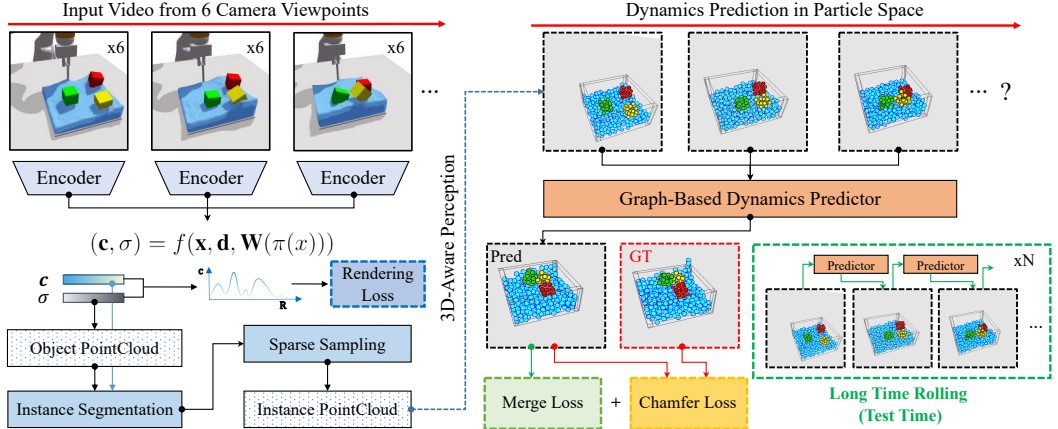

Figure 2: **Overview of 3D Visual Intuitive Physics (3D-IntPhys).** Our model consists of two major components: **Left:** The perception module maps the visual observations into implicit neural representations of the environment. We then subsample from the reconstructed implicit volume to obtain a particle representation of the environment. **Right:** The dynamics module, instantiated as graph neural networks, models the interaction within and between the objects and predicts the evolution of the particle set.

works by Bear et al. (2021) have shown that only methods that use 3D view-invariant representations can pave the way toward human-level physics dynamics prediction in diverse scenarios.

Researchers have attempted to learn object motion in 3D (Tung et al., 2020; Xu et al., 2020; Manuelli et al., 2020; Li et al., 2021b). Tung et al. (2020) and Xu et al. (2020) use object-centric volumetric representations inferred from RGB-D to predict object motion, yet, these volumetric approaches have much higher computation costs than 2D methods due to the 4D representation bottleneck, which hinders them from scaling up to more complex scenes. Manuelli et al. (2020) use self-supervised 3D keypoints and Driess et al. (2022a;b) use implicit representations to model multi-object dynamics but cannot handle objects with high degrees of freedom like fluid and granular materials. Li et al. (2021b) use neural implicit representation to reduce the potential computational cost, yet the works have not shown how the approach can generalize to unseen scenarios. Our works aim to solve the tasks of learning generalizable object dynamics in 3D by combining the generalization strength of input-feature-conditioned implicit representation and point-based dynamics models.

**Point-based dynamics models.** Existing works in point- and mesh-based dynamics models (Li et al., 2019a; Mrowca et al., 2018; Ummenhofer et al., 2019; Sanchez-Gonzalez et al., 2020; Pfaff et al., 2021) have shown impressive results in predicting the dynamics of rigid objects, fluid (Li et al., 2019a; Mrowca et al., 2018; Ummenhofer et al., 2019; Sanchez-Gonzalez et al., 2020; Allen et al., 2022; de Avila Belbute-Peres et al., 2020), deformable objects (Li et al., 2019a; Mrowca et al., 2018; Sanchez-Gonzalez et al., 2020), and clothes (Pfaff et al., 2021; Lin et al., 2021). Most works require access to full 3D states of the points during training and testing, yet, such information is usually not accessible in a real-world setup. Li et al. (2020) learn a visual frontend to infer 3D point states from images, but still require 3D point states and trajectories during training time. Shi et al. (2022) propose to learn point dynamics directly from vision, but they only consider elasto-plastic objects consisting of homogeneous materials. How to learn about 3D point states and their motion from raw pixels remain a question. Our paper tries to build the link from pixels to points using recent advances in unsupervised 3D inference from images using NeRF (Mildenhall et al., 2020; Yu et al., 2021).

## 3 METHODS

We present 3D Visual Intuitive Physics (3D-IntPhys), a model that learns to simulate physical events from unlabeled images (Figure 2). 3D-IntPhys contains a perception module that transforms visual observations into a 3D point cloud that captures the object geometries (Section 3.1) and a point-based simulator that learns to simulate the rollout trajectories of the points (Section 3.2). The design choice of learning physics simulation in a 3D-point representation space enables stronger simulation performance and generalization ability. The performance gain mainly comes from the fact that describing/learning objects' motion and interactions in 3D are easier compared to doing so in 2D since objects live and move persistently in the 3D space. 3D-IntPhys also supports better

generalization ability since its neural architecture explicitly models how local geometries of two objects/parts interact, and these local geometries and interactions can be shared across different and novel object combinations.

Although 3D-IntPhys learns to simulate in a 3D representation space, we show it can learn without any 3D supervision such as dense point trajectories as in previous work (Sanchez-Gonzalez et al., 2020; Li et al., 2019a). Dense point trajectories are hard and sometimes impossible to obtain in the real world, e.g., capturing the trajectories of each water point. 3D-IntPhys does not require such 3D supervision and can simply learn by observing videos of the scene evolution.

### 3.1 2D-TO-3D PERCEPTION MODULE

Given a static scene, the perception module learns to transform one or a few posed RGB images, $\mathbf{I} = \{(I_i, \pi_i)|i \in \{1, 2, \cdots, N_v\}\}$, taken from $N_v$ different views, into a 3D point cloud representation of the scene, $\mathbf{X}$. We train the model in an unsupervised manner through view reconstruction, using a dataset consisting of $N_t$ videos, where each video has $N_f$ frames, and each frame contains images taken from $N_v$ viewpoints.

**Neural Radiance Field (NeRF).** NeRF (Mildenhall et al., 2020) learns to reconstruct a volumetric radiance field of a scene from unlabeled multi-view images. After training, the model learns to predict the RGB color $\mathbf{c}$ and the corresponding density $\sigma$ of a query 3D point $\mathbf{x} \in \mathbb{R}^3$ from the viewing direction $\mathbf{d} \in \mathbb{R}^3$ with a function $(\mathbf{c}, \sigma) = f(\mathbf{x}, \mathbf{d})$. We can formulate a camera ray as $\mathbf{r}(t) = \mathbf{o} + t\mathbf{d}$, where $\mathbf{o} \in \mathbb{R}^3$ is the origin of the ray. The volumetric radiance field can then be rendered into a 2D image via $\hat{\mathbf{C}}(\mathbf{r}) = \int_{t_n}^{t_f} T(t)\sigma(t)\mathbf{c}(t)dt$, where $T(t) = \exp(-\int_{t_n}^{t} \sigma(s)ds)$ handles occlusion. The rendering range is controlled by the depths of the near and far plane (i.e., $t_n$ and $t_f$). We can train NeRF through view prediction by:

$$\mathcal{L} = \sum_{\mathbf{r} \in \mathcal{R}(\mathbf{P})} \|\hat{\mathbf{C}}(\mathbf{r}) - \mathbf{C}(\mathbf{r})\|, \tag{1}$$

where $\mathcal{R}(\mathbf{p})$ is the set of camera rays sampled from target camera pose $\mathbf{p}$.

**Image-conditioned NeRF.** To infer the NeRF function from an image, previous work proposed to encode the input image into a vector, with a CNN encoder, as a conditioning input to the target NeRF function (Li et al., 2021b). We found this type of architecture is in general hard to train and does not generalize well. Instead, we adopt pixelNeRF (Yu et al., 2021), which conditions NeRF rendering with local features, as opposed to global features. Given an image $I$ in a scene, pixelNeRF first extracts a feature volume using a CNN encoder $\mathbf{W} = E(I)$. For a point $\mathbf{x}$ in the world coordinate, we retrieve its feature vector by projecting it onto the image plane, so that we can get the feature vector $\mathbf{W}(\pi(\mathbf{x}))$. PixelNeRF combines the feature vector together with the 3D position of that point and predict the RGB color and density information:

$$\mathbf{V}(\mathbf{x}) = (\mathbf{c}, \sigma) = f(\mathbf{x}, \mathbf{d}, \mathbf{W}(\pi(\mathbf{x}))). \tag{2}$$

PixelNeRF can also incorporate multiple views to improve predictions when more input views are available; this will help decrease ambiguity caused by occlusion and will greatly help us get better visual perceptions of the target scene.

**3D point representation from pixelNeRF.** From a few posed RGB images, $\mathbf{I}$, of a scene $s$, we infer a set of points for $O_s$ target object (such as fluid, cube) in the scene. We achieve this by first sampling a set of points according to the predicted occupancy measure, then clustering the points into objects using object segmentations. We found that sampling with low resolution will hurt the quality of the rendered point cloud to generate objects with inaccurate shapes, while sampling with high resolution will increase the computation for training the dynamics model since the input size increases. To speed up training while maintaining the quality of the reconstructed point cloud, we first infer the points with higher resolution and do sparse sampling of each point cloud using FPS (Farthest Point Sampling) (Eldar et al., 1997). Next, we cluster the inferred points into objects according to object segmentation masks. Since solving object segmentation in general is not the main focus of this paper, we resort to using the color information to obtain the masks.

### 3.2 POINT-BASED DYNAMICS SIMULATOR

Given the point representation at the current time step, $\mathbf{X}_t$, the dynamics simulator predicts the points' evolution $T$ steps in the future, $\{\mathbf{X}_{t+1}, \mathbf{X}_{t+2}, \cdots \mathbf{X}_{t+T}\}$, using graph-based networks (Sanchez-Gonzalez et al., 2020; Li et al., 2019a). We first form a graph $(V, E)$ based on the distance between

points. If the distance between two points is smaller than a threshold $\delta$, we include an edge between these two points. Each vertex $v_i = (\dot{x}_i, a_i^v) \in V$ contains the velocity of the point, $\dot{x}_i$, and point attributes, $a_i^v$, to indicate the point's type. For each relation, $(i, j) \in E$, we have its associated relation attribute $a_{ij}^e$, indicating the types of relation and the relative distance between the connected points.

**Spatial message passing and propagation.** At time step $t$, we can do message passing to update the points, $v_i \in V$, and relation representations, $(i, j) \in E$, in the graph:

$$g_{ij,t} = Q_e(v_{i,t}, v_{j,t}, a_{ij}^e) \qquad\qquad (i, j) \in E \qquad\qquad (3)$$

$$h_{i,t} = Q_v(v_{i,t}, \sum\nolimits_{k \in \{j | (i,j) \in E\}} g_{ik,t}) \qquad v_i \in V \qquad\qquad (4)$$

where $Q_v$ and $Q_e$ are encoders for vertices and relations respectively. Please refer to Battaglia et al. (2018) for more details. Though this kind of message passing can help with updating representation, it can only share one-hop information in each step, limiting its performance on instantaneous passing of forces. To improve long-range instantaneous effect propagation, we use multi-step message propagation as in Li et al. (2019b;a). Starting from propagation step 0 with an initialization step where $h_{i,t}^0 = h_{i,t}$ for all vertices and $g_{ij,t}^0 = g_{ij,t}$ for all edges, we do $L$ steps of propagation by

$$\text{Step } l \in \{1, 2, ..., L\}: \qquad g_{ij,t}^l = P_e(g_{ij,t}^{l-1}, h_{i,t}^{l-1}, h_{j,t}^{l-1}) \qquad (i, j) \in E \qquad (5)$$

$$h_{i,t}^l = P_v(h_{i,t}^{l-1}, \sum\nolimits_{k \in \{j | (i,j) \in E\}} g_{ik,t}^l) \quad v_i \in V \qquad (6)$$

where $P_e, P_v$ are propagation functions of nodes and edges, respectively, and $g_{ij,t}^l$ is the effect of relation $(i, j)$ in propagation step $l$. $h_{i,t}^l$ is the hidden states for each point in the propagation process. Finally, we have the predicted states of points at time step $t + 1$ after $L$ steps of propagation:

$$\hat{v}_{i,t+1} = f_{\text{pred}}(h_{i,t}^L). \qquad\qquad (7)$$

**Environments.** We assume that the surrounding environment (e.g., the table) is known and the robot/tool/container are of known shape and fully actuated, where the model has access to their complete 3D state information. We convert the full 3D states into points through sampling on the 3D meshes and include these points in the prediction of the graph-based dynamics.

**Fluids, rigid bodies, and granular materials.** We distinguish different materials by using different point attributes $a_i^v$. We also set different relation attributes $a_{ij}^e$ in Equation 3 to distinguish different interaction (e.g., Rigid-Fluids, Fluids-Fluids, Granular-Pusher). For rigid objects, to ensure the object shapes remain consistent throughout the rollout predictions, we add a differentiable rigid constraint in the prediction head following Li et al. (2019a).

**Training dynamics model without point-level correspondence.** Since our perception model parses each RGB image into object-centric point clouds independently, there does not exist an explicit one-to-one correspondence for points across frames. To handle this, we measure the Chamfer distance between the prediction $\hat{\mathbf{X}}_t = (\hat{V}_t, \hat{E}_t)$ from the dynamics network and the inferred point state $\mathbf{X}_t = (V_t, E_t)$ from the perception module and treat it as the objective function. The Chamfer distance between two point cloud $\hat{V}$ and $V$ is defined as:

$$L_c(\hat{V}, V) = \frac{1}{\|\hat{V}\|} \sum_{x \in \hat{V}} \min_{y \in V} \|x - y\|_2^2 + \frac{1}{\|V\|} \sum_{x \in V} \min_{y \in \hat{V}} \|x - y\|_2^2. \qquad (8)$$

We found that training the model with Chamfer distance in dense scenes with granular materials will often lead to predictions with unevenly distributed points where some points stick too close to each other. To alleviate this issue, we further introduce a spacing loss $L_s$, which penalizes the gated distance (gated by $d_{\min}$) of nearest neighbor of each point to ensure enough space between points:

$$L_s(\hat{V}) = \sum\nolimits_{v \in \hat{V}} (\text{ReLU}(d_{\min} - \min_{v' \in \{\hat{V} \setminus v\}} \|v' - v\|_2^2))^2. \qquad (9)$$

The one-step prediction loss $L_{dy}$ for training the dynamics model is $L_c(\hat{V}, V) + \sigma L_s(\hat{V})$ where $\sigma$ reweights the second loss. To improve long-term rollout accuracy, we train the model with two-step predictions using the first predicted state as input and feed it back into the model to generate the second predicted state. With the two-step loss, the model becomes more robust to errors generated from its own prediction. Finally, the $L_{dy}$ losses for all rolling steps are summed up to get the final loss for this trajectory. More implementation details are included in the supplementary material.

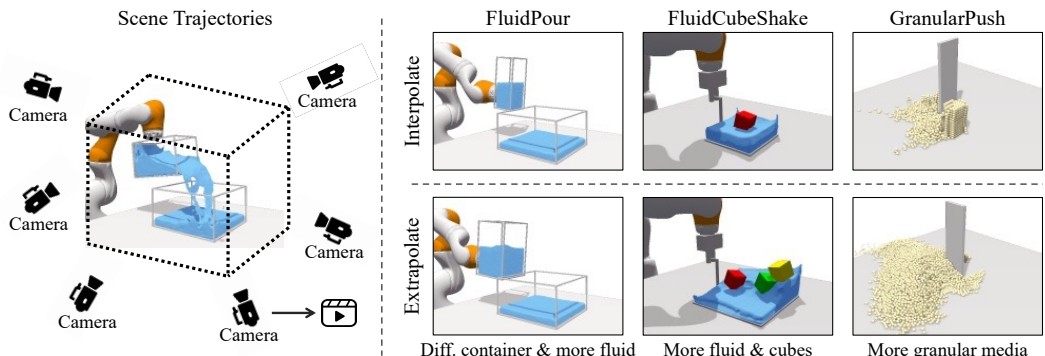

Figure 3: **Data Collection and Evaluation Setups. Left:** We collect multi-view videos of the environment from six cameras. **Right:** We consider a diverse set of evaluating environments involving fluids, rigid objects, granular materials, and their interactions with the fully-actuated container and the environment. We evaluate the learned visual intuitive physics model on both the interpolated settings (i.e., seen environment but with different action sequences) and extrapolated settings (i.e., unseen environment with different amounts of fluids, cubes, granular pieces, and containers of different sizes).

## 4 EXPERIMENTS

The experiment section aims to answer the following three questions. (1) How well can the visual inference module capture the content of the environment (i.e., can we use the learned representations to reconstruct the scene)? (2) How well does the proposed framework perform in scenes with objects of complicated physical properties (e.g., fluids, rigid and granular objects) compared to baselines without explicit 3D representations? (3) How well do the models generalize in extrapolate scenarios?

**Datasets.** We generated three simulated datasets using the physics simulator Nvidia FleX (Macklin et al., 2014). Each of the datasets represents one specific kind of manipulation scenario, where a robot arm interacts with rigid, fluid, and granular objects (Figure 3). For each of the three scenarios, we apply randomized input actions and change some properties of objects in the scene, e.g., the shape of the container, the amount of water, and the color/number of cubes, to make it diverse. To test the generalization capability of the trained model, we design extrapolated datasets where the data is generated from an extrapolated set of parameters outside the training distribution.

**a) FluidPour.** This scenario contains a fully-actuated cup pouring fluid into a container. We design the extrapolate dataset to have a larger container, more quantity of fluid, and different pouring actions.

**b) FluidCubeShake.** This scenario contains a fully-actuated container that moves on top of a table. Inside the container are fluids and cubes with diverse colors. We design the extrapolate dataset to have different container shapes, number of cubes, cube colors, and different shaking actions.

**c) GranularPush.** This environment contains a fully-actuated board pushing a pile of granular pieces. We design the extrapolate dataset to have a larger quantity of granular objects in the scene than the model has ever seen during training.

**Baselines.** We compare our method with two baselines, NeRF-dy (Li et al., 2021b) and autoencoder (AE) (similar to GQN (Eslami et al., 2018) augmented with a latent-space dynamics model). NeRF-dy is a 3D-aware framework that also learns intuitive physics from multi-view videos. Yet, instead of learning the object dynamics with explicit and compositional 3D representations, the model learns dynamics models with implicit 3D representations in the form of a single latent vector. We also compare our method with an autoencoder-based reconstruction model (AE) (Eslami et al., 2018) that can perform novel-view synthesis but is worse at handling 3D transformations than neural implicit representations. AE first learns scene representations through per-frame image reconstruction, and then it learns a dynamics model on top of the learned latent representations. All methods take RGB images and camera parameters as inputs. To incorporate object-level information, we perform color-based segmentation to obtain object masks as additional inputs to the baselines.

**Implementation details.** We train and test the perception module with 6 camera views. To obtain a set of points from the learned perception module, we sample points on a $40 \times 40 \times 40$ grid from an area of 55cm $\times$ 55cm $\times$ 55cm at the center of the table for FluidPour, 63cm $\times$ 63cm $\times$ 63cm with $40 \times 40 \times 40$ grids for FluidCubeShake, and on a $70 \times 70 \times 70$ grid area for GranularPush.

| | | FluidPour | | FluidCubeShake | | GranularPush | |
|---|---|---|---|---|---|---|---|
| Metrics | Model | InD | OoD | InD | OoD | InD | OoD |
| MSE(↓) | AE | 451.03 | 542.86 | 869.3 | 1727.55 | 562.06 | 1537.2 |
| | NeRF-dy | 202.95 | 317.27 | 527.46 | 1585.97 | 481.95 | 1020.0 |
| | Ours | **111.66** | **124.33** | **66.52** | **81.38** | **147.97** | **646.85** |
| SSIM(↑) | AE | 0.86 | 0.84 | 0.71 | 0.86 | 0.81 | 0.62 |
| | NeRF-dy | 0.89 | 0.86 | 0.73 | 0.65 | 0.81 | 0.61 |
| | Ours | **0.90** | **0.89** | **0.94** | **0.93** | **0.89** | **0.69** |

Table 1: **Quantitative Results of the Perception Module.** We compare our method with autoencoder (AE) and NeRF-dy (Li et al., 2021b) with additional instance masks based on color. We measure the quality of rendered images by computing the Mean Squared Error (MSE) and Structural Similarity Index Measure (SSIM) compared to the ground truth. InD stands for in-distribution tests, and OoD stands for out-of-distribution tests.

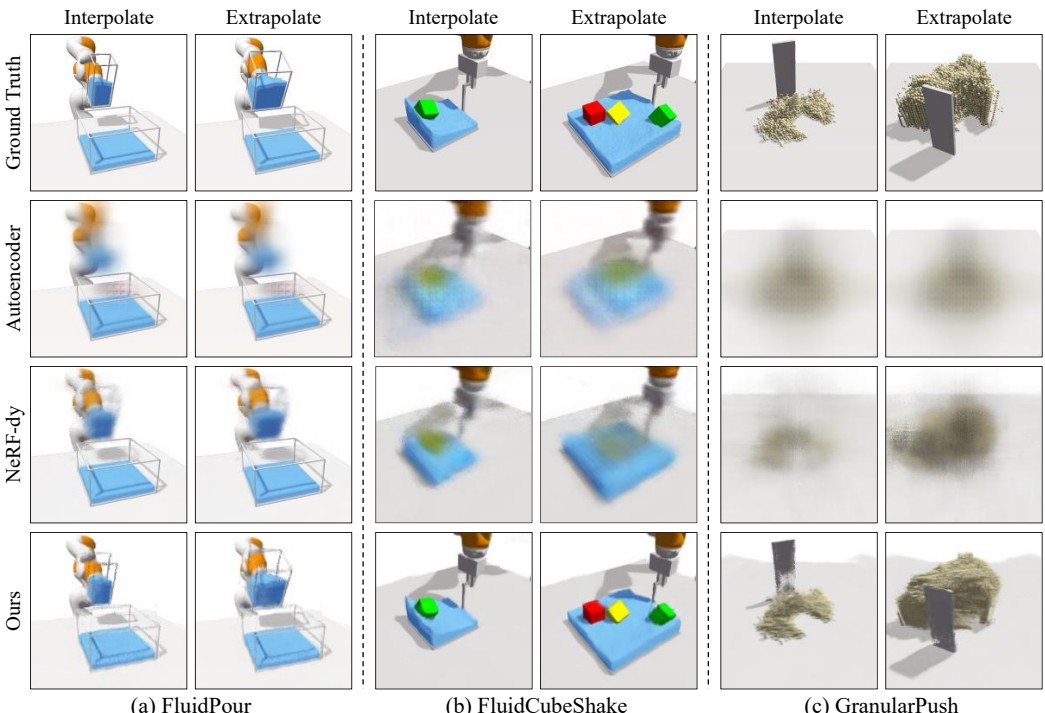

(a) FluidPour          (b) FluidCubeShake          (c) GranularPush

Figure 4: **Qualitative Reconstruction Results of the Perception Module.** The images generated by our method contain more visual details and are much better aligned with the ground truth. Our model is much better at handling large scene variations than NeRF-dy, especially in extrapolate settings.

We evaluate and include points with a density (measured by the occupancy in the predicted neural radiance fields) larger than $0.99$. We subsample the inferred points with FPS with a ratio of around $5\%$ for FluidPour and $10\%$ for FluidCubeShake and GranularPush. We currently determine the sampling rate by manually picking the minimum rate that yields a reasonable point cloud describing the object shapes. The threshold $d_{\min}$ is set to $0.08$ and the weighting $\sigma$ is set to $10$. We select the distance so that each point will have on average 20 to 30 neighbors. Details for data generation parameters, model architecture, training schema, and more data samples are included in the supplementary material.

## 4.1 Image Reconstruction From Learned Scene Representations

We test how well the perception modules capture scene information by evaluating the visual front-end of all models on their ability to reconstruct the observed scene from the inferred representations. We measure the difference between the reconstructed and ground truth images with Mean Squared Error (MSE) and Structural Similarity (SSIM) in pixel level (Table 1). Our perception module outperforms all baselines in all three environments. The performance gap is exaggerated in extrapolate settings, especially in scenarios that involve complex interactions between rigid and deformable materials (see Figure 4 for qualitative comparisons).

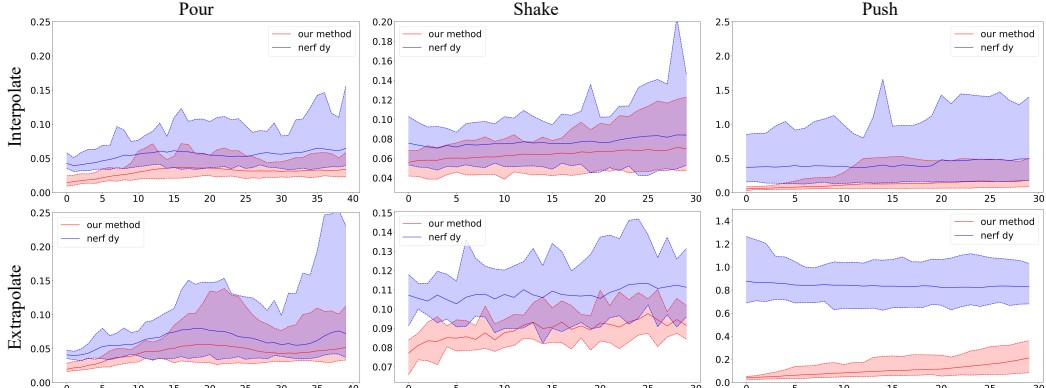

Figure 5: **Quantitative Results of the Dynamics Module.** This figure compares our method and NeRF-dy (Li et al., 2021b) on their long-horizon open-loop future prediction performance. The loss is measured as the Chamfer distance between the predicted particle set evolution and the actual future. Our method outperforms the baseline in both interpolate and extrapolate settings, showing the benefits of explicit 3D modeling.

## 4.2 LEARNED VISUAL DYNAMICS ON IN-DISTRIBUTION HELD-OUT SCENES

Next, we compare long-term rollouts in the 3D space. We evaluate the models using the Chamfer distance between the predicted point cloud and the ground truth. For NeRF-dy, we decode the predicted rollouts latent vectors into the point cloud with the learned NeRF decoder. We exclude the comparison with AE since it is unclear how to decode the learned representations into point clouds. We show quantitative comparison in Figure 5 and qualitative results in Figure 6. 3D-IntPhys can learn reasonable scene dynamics in all scenarios and significantly outperforms NeRF-dy. While NeRF-dy can learn relatively reasonable movements of fluids, it fails to learn complex dynamics such as the floating cube and the morphing of the granular materials. The results suggest that the proposed explicit 3D point-based representations are critical to learning complex multi-material dynamics.

## 4.3 GENERALIZATION ON OUT-OF-DISTRIBUTION SCENES

To test the generalization ability of the models, we introduce extrapolate settings of all of the three scenarios. See "Extrapolate" results in Table 1, Figure 4, 5, and 6. The proposed 3D-IntPhys generalizes well to extrapolate settings both at the visual perception stage and the dynamics prediction stage, whereas NeRF-dy and autoencoder both fail at generalizing under extrapolate settings. For example, in **FluidShake**, both baselines cannot capture the number and the color of the rigid cubes (Figure 4). And in **GranularPush**, both baselines fail to capture the distributions of the granular materials. NeRF-dy performs much worse on extrapolation scenes compared to in-distribution scenes, suggesting that incorporating 3D information in an explicit way, as opposed to implicit, is much better at capturing the structure of the underlying environment, thus leading to better generalization. We further test our model on completely unseen changes to the environment – in the **GranularPush** environment, we extend the width of the pusher by a factor of 2 and 5. Though the stretched pusher has never shown in the training data, our model can make reasonable pushing predictions (see Fig 7).

## 5 CONCLUSIONS

In this work, we propose a 3D-aware and compositional framework, 3D-IntPhys, to learn intuitive physics from unlabeled visual inputs. Our framework can work on complex scenes involving fluid, rigid objects, and granular materials, and generalize to unseen scenes with containers of different sizes, more objects, or larger quantities of fluids and granular pieces. We show the proposed model outperforms baselines by a large margin, highlighting the importance of learning dynamics models in an explicit 3D representations space. One major limitation of our work is the assumption of the access to object masks. Although the masks do not have to be perfect and there has been significant progress on (self-)supervised object segmentation in recent years, it can still be hard to obtain these masks in more complicated real-world settings. An exciting future direction is to learn the segmentation and material properties jointly with the dynamics from the observation data.

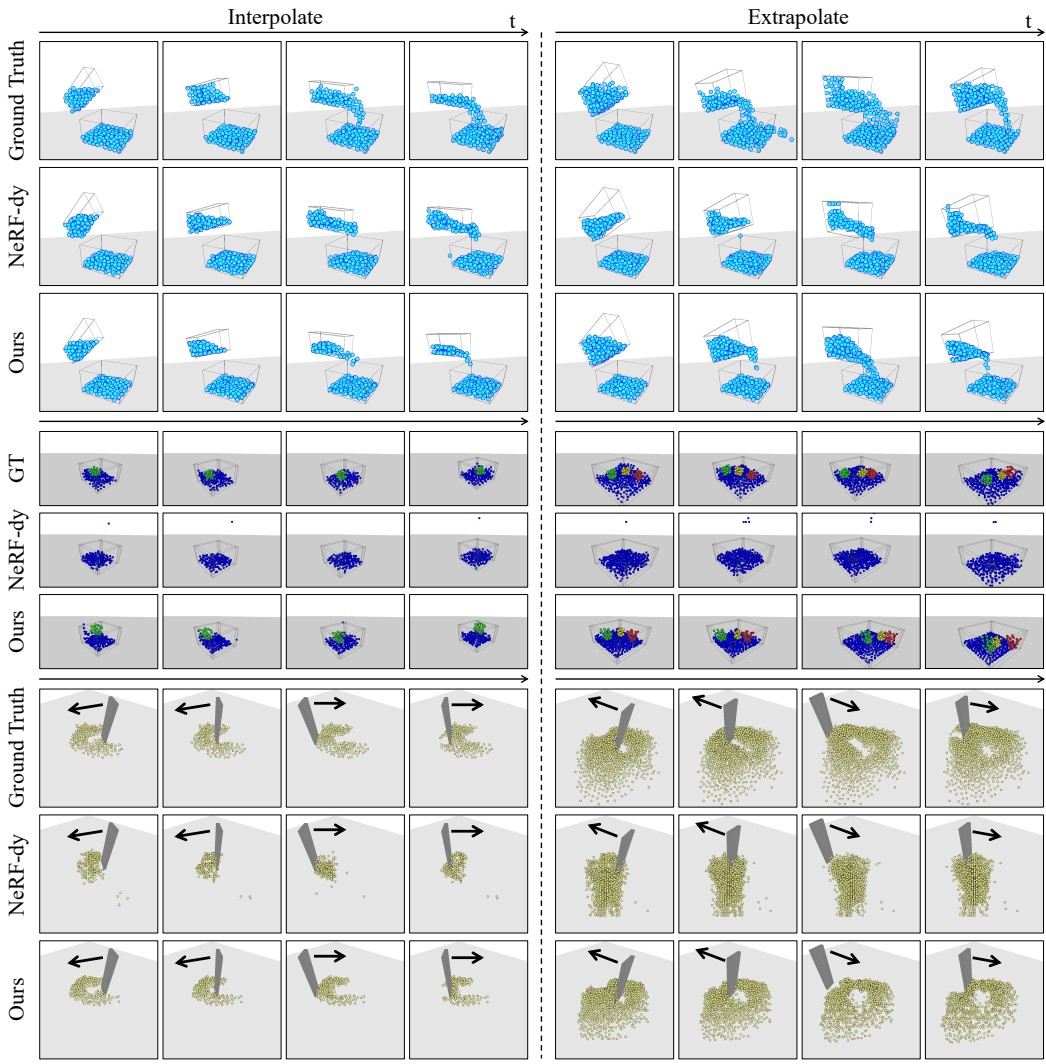

Figure 6: **Qualitative Results of the Dynamics Module on Future Prediction.** Here we visualize our model's predicted future evolution of the particle set as compared with the NeRF-dy (Li et al., 2021b) baseline in both interpolate and extrapolate settings. Our method correctly identifies the shape/distribution of the fluids, rigid objects, and granular pieces with much better accuracy than NeRF-dy. The future evolution predicted by our method also matches the ground truth much better and produces reasonable results even in extrapolate settings.

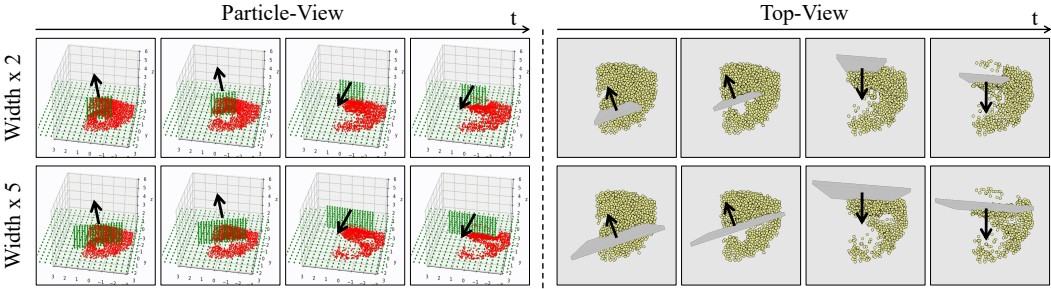

Figure 7: **Strong Generalization Ability of the Dynamics Module to Wider Pushers.** We evaluate our dynamics model on unseen width of pushers in **GranularPush** environment. The left part shows in point space where red indicates granular materials, green shows table and pusher, and the arrow shows how the pusher is about to move. The right part shows from top view of rendering results.

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

# A    ADDITIONAL RESULTS

To better understand the performance of our framework visually, we prepare test time rollouts of our framework as well as those of various baselines in the supplementary video.

## A.1    ABLATION STUDY

We find that training the model with Chamfer distance in dense scenes with granular materials will often lead to predictions with unevenly distributed points where some points stick too close to each other. To alleviate the issue, we introduce the spacing loss to penalize the distance between these points. We set the threshold of penalty $d_{min}$ to be 0.08 and the loss weight $\sigma$ to be 10. We find that spacing loss can help improve the performance of the dynamics learner especially under extrapolate settings, as shown in Figure 8. We provide qualitative results in the supplementary video.

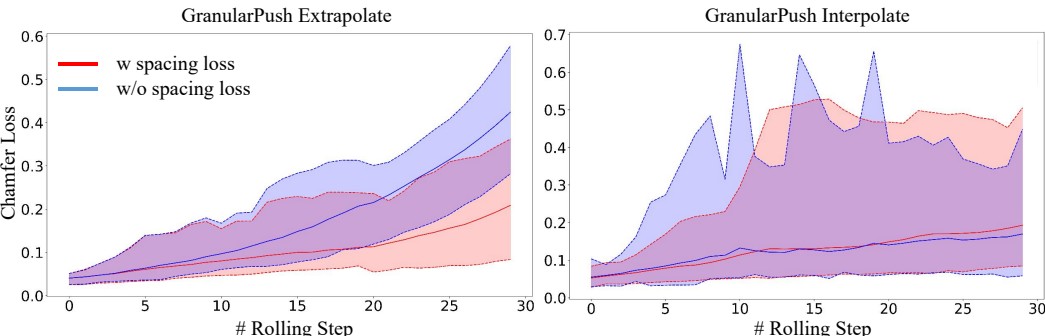

Figure 8: **Ablation Study on the Spacing Loss.** Training dynamics models in the GranularPush scenario with spacing loss results in better rolling prediction. The performance gap is even more substantial in the extrapolate setting.

# B    IMPLEMENTATION DETAILS

## B.1    DATASET GENERATION

Our datasets are generated by the NVIDIA Flex simulator (Macklin et al., 2014). Each of the three scenarios (Pour, Shake and Push) has 500 videos of trajectories taken from 6 views, with each trajectory consisting of 300 frames. We manually select the 6 views with reasonable coverage of the tabletop space to minimize the occlusion. We have included the camera parameters in the OpenGL format in the supplementary zip file (camera_viewmatrix.txt). The 500 trials are generated from five different sets of environmental parameters, detailed in Table 3. We take one set of parameters that are outside the training distribution as the **extrapolate** dataset for evaluating model generalization. For the rest of the four settings, we randomly split them into train and test sets with a ratio of 0.8.

Next, we provide more details for each scenario:

- In the FluidPour environment, we randomly initialize the position of the upper container and then generate random back-and-forth actions by tilting the container. The action space is then the position and tilting angle of the upper container.
- In FluidCubeShake, we also randomly initialize the position of the container and the cubes inside the container. We then generate random but smooth action sequences moving the container in the 2D plane. The action space is then the x-y location of the container.
- In GranularPush, we randomly initialize the position of the granular pile. Then, for each push, we randomly generate the starting and ending positions of the pusher and move the pusher along the straight line with an angle perpendicular to the pushing direction. The action space is a four-number tuple stating the starting and ending position on the 2D plane.

The following table shows the moving range of the robot arms in the FluidPour and FluidCubeShake environments after normalizing the robot into a size that is the same as in the real world (unit:

|  | X-Range | Y-Range | Z-Range |
|---|---|---|---|
| FluidPour | [-29.11, -12.66] | [42.00, 60.00] | [-7.78, 7.78] |
| FluidCubeShake | [-3.25, 42.25] | [19.25, 19.25] | [-24.50, 24.00] |

Table 2: **Robot Action Space(centimeters):** we show the range the robot arms can move in the FluidPour and FluidCubeShake environments.

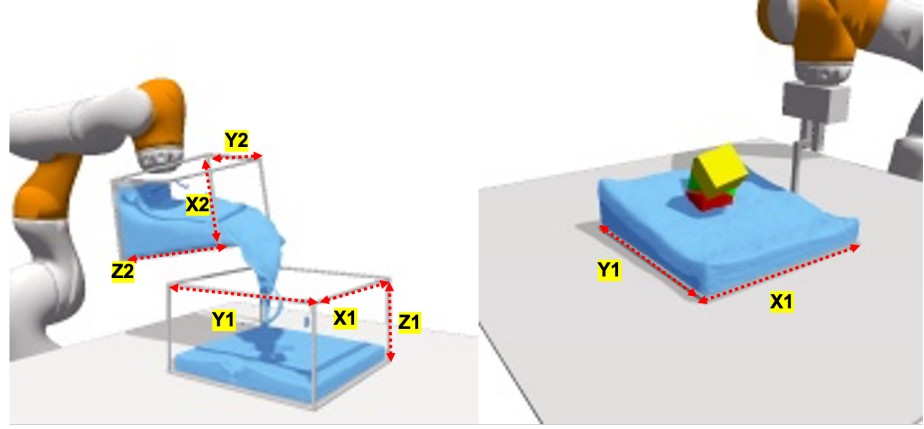

Figure 9: **Illustration of the Environment Settings.** In the FluidPour scenario, a robot arm holds a container and tries to pour some fluid into another container. In the FluidShake scenario, a robot moves a container with some fluid and cubes. We show the parameters for the container shape referred in Table 3.

| SceneName | Params | Env1 | Env2 | Env3 | Env4 | Extrapolate |
|---|---|---|---|---|---|---|
| FluidPour | X2 | 0.53 | 0.53 | 0.81 | 0.81 | 0.81 |
|  | Y2 | 0.53 | 0.81 | 0.53 | 0.81 | 0.81 |
|  | Z2 | 1.24 | 1.24 | 1.24 | 1.24 | 1.24 |
|  | X1 | 1.35 | 1.35 | 1.35 | 1.35 | 1.35 |
|  | Y1 | 1.35 | 1.35 | 1.35 | 1.35 | 1.35 |
|  | Z1 | 0.74 | 0.74 | 0.74 | 0.74 | 0.74 |
|  | AmountofWater | 5125 | 5125 | 6125 | 5375 | 7625 |
| FluidCubeShake | X1 | 0.88 | 0.88 | 1.32 | 1.32 | 1.32 |
|  | Y1 | 0.88 | 1.32 | 0.88 | 1.32 | 1.32 |
|  | CubeNumber | 1 | 1 | 2 | 2 | 3 |
|  | Water | 2173 | 3322 | 3322 | 4858 | 4983 |
| GranularPush | GranularNumber | 2197 | 4032 | 5832 | 9261 | 12167 |

Table 3: **Scene Parameters for Generating the Interpolate and Extrapolate Datasets.** We generate the datasets by varying the shape of container, amount of water, number of cubes, and quantity of the granular material. $Z_i, X_i, Y_i$ are the height, width, and depth for a container $i$. Please refer to Figure 9 for more details.

centimeters). For GranularPush, the pusher is moving over the entire table; we ignore the specific number in this environment as we do not have robot arms as a reference.

**Additional dataset samples.** We show samples from the FluidPour, FluidCubeShake and GranularPush dataset in Figure 10, 11 and 12, respectively. Note that all trajectories for the extrapolate settings are used only for testing and will not show up during the training process. We include more samples from the dataset in the video format in the supplementary video.

## B.2 MODEL ARCHITECTURE

**Image-conditional NeRF.** We follow the architectural design by Yu et al. (2021). For the feature encoder, we employ a ResNet-34 backbone to extract features. We use the output layers prior to the first four pooling layers, upsampling them using bilinear interpolation to the same size, and

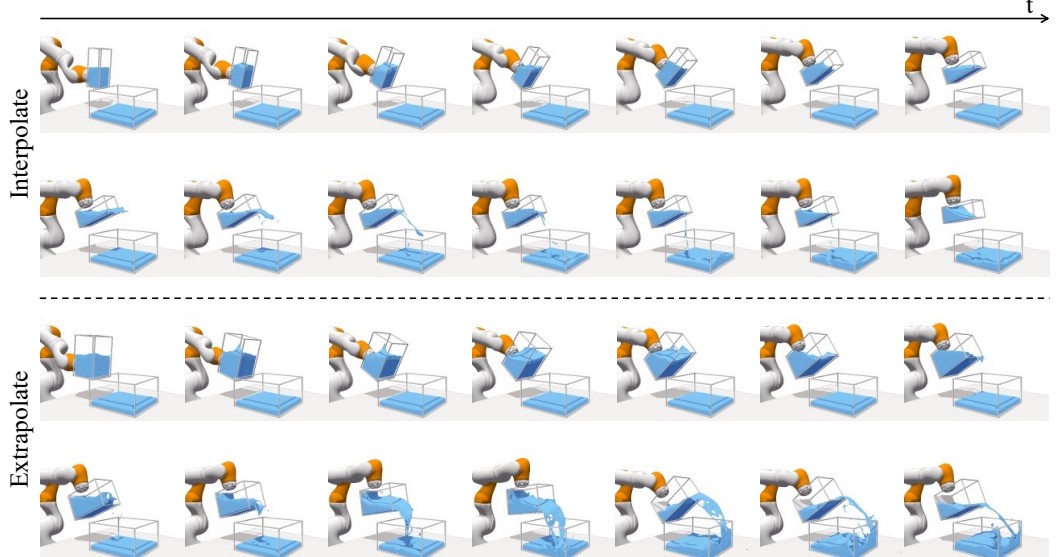

Figure 10: **Samples from FluidPour Dataset.** We show sequences of frames over time with an interval of 20 frames. The sequences above the dashed line are for **interpolate** data, and the bottom images illustrate the **extrapolate** data.

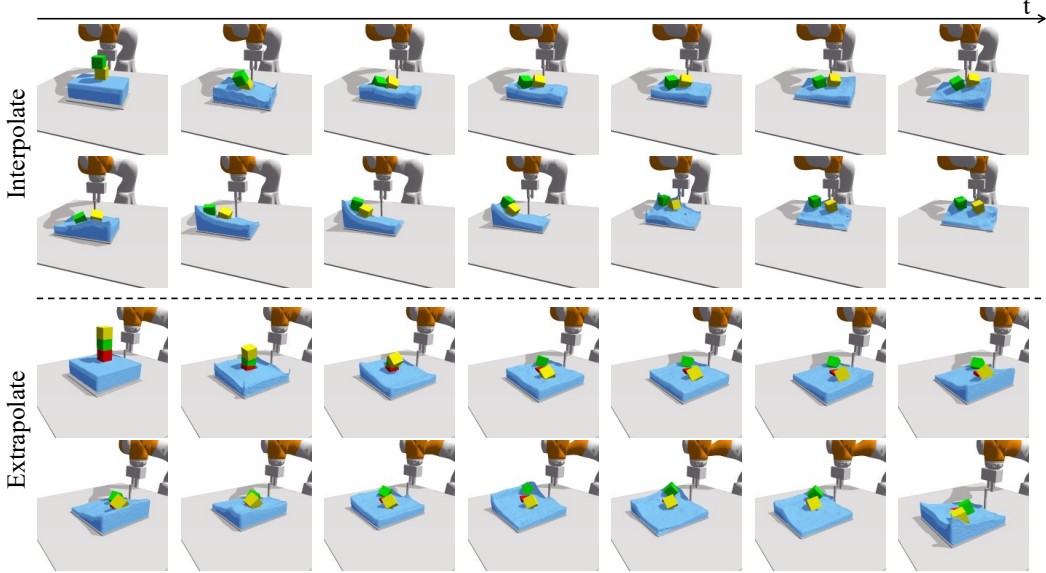

Figure 11: **Samples from FluidCubeShake Dataset.** We show sequences of frames over time with an interval of 20 frames. The sequences above the dashed line are for **interpolate** data, and the bottom images illustrate the **extrapolate** data.

then concatenating these four feature maps. We initialize the weight of the feature extractor of the scene using ImageNet pre-trained weight. For the NeRF function $f$, We use fully-connected ResNet architecture with 5 ResNet blocks with a width of 512.

**Dynamics predictor.** For the edge and vertice encoders, $Q_e$ and $Q_v$, we use 3-layer fully-connected networks activated by the ReLU function with 150 hidden units. For the propagators, $P_e$ and $P_v$, we use a 1-layer fully-connected network followed by ReLU activation. The output dimension of the linear layer is 150.

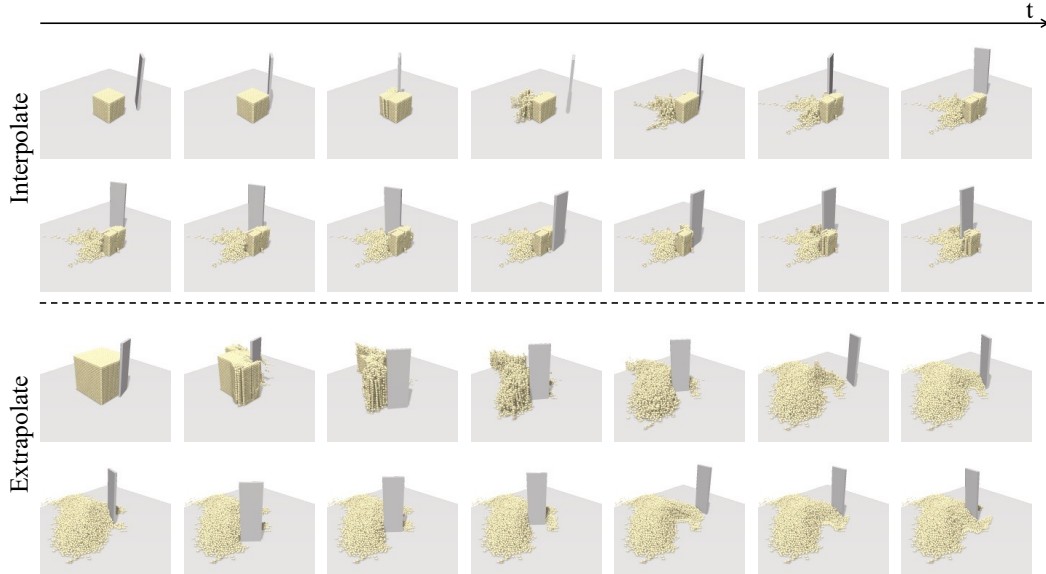

Figure 12: **Samples from GranularPush Dataset.** We show sequences of frames over time with an interval of 20 frames. The sequences above the dashed line are for **interpolate** data, and the bottom images illustrate the **extrapolate** data.

**Sampling 3D points from the trained visual perception module.** We sample points on a $40 \times 40 \times 40$ grid from an area of $55cm \times 55cm \times 55cm$ and $63cm \times 63cm \times 63cm$ at the center of the table for FluidPour and FluidCubeShake respectively, and on a $70 \times 70 \times 70$ grid from an area of $6cm \times 6cm \times 6cm$ for GranularPush. We evaluate and include points with a density (measured by the occupancy in the predicted neural radiance fields) larger than $0.99$. To reduce the total number of points, we subsample the inferred points with FPS with a ratio of $5\%$ for FluidPour and $10\%$ for FluidCubeShake and GranularPush.

**Graph building.** We set the neighbour distance threshold $\delta$ to be $0.2, 0.15, 0.15$ for FluidPour, FluidCubeShake and GranularPush respectively. We select the threshold so that each point will have on average 20 30 neighbors. Since, in FluidPour, we sample the points with lower density $2000$points$/m^2$, we use a larger threshold for this scenario. For FluidShape and GranularPush, since the density is around $3000$ points$/m^2$, we cut down the number by $25\%$.

We found that if the threshold is too small, the performance will degrade significantly since each particle will only receive messages from a few neighbors (and miss out on the larger context). On the other hand, setting the threshold too large will cause the training time to increase since the graph will have more edges. We found that setting the threshold around the right scale generally leads to more effective training of a reasonable dynamics network.

### B.3 TRAINING DETAILS

The models are implemented in PyTorch. We train the perception module using Adam optimizer with a learning rate of $1e-4$, and we reduce the learning rate by $80\%$ when the performance on the validation set has stopped improving for 3 epochs. To compute the rendering loss when training the perception module, we sample 64 points through each ray in the scene and set the ray-batch size of the NeRF query function $f$ to be $1024 \times 32$. Training the perception module on a single scenario takes around 5 hours on one RTX-3090.

We train the dynamics simulator using Adam optimizer with a learning rate of $1e-4$, and we reduce the learning rate by $80\%$ when the performance on the validation set has stopped improving for 3 epochs. The batch size is set to 4. We train the model for 20, 30, and 40 epochs for FluidPour, FluidCubeShake, and GranularPush, respectively. It takes around $10 \sim 15$ hours to train the dynamics model in one environment on one single RTX-3090.

### B.4 GRAPH-BASED DYNAMICS MODEL WITHOUT PARTICLE-LEVEL CORRESPONDENCE

The velocity of an object provides critical information on how the object will move in the future, yet, we do not have access to such information when tracking the object is impossible. As described in Section 3.2, the attributes $a_i^v$ of a vertex $v_i$ in the built graph consists of (1) velocity of this point in the past frames and (2) attributes of the point (rigid, fluid, granular). To get the velocity of a vertex $v$, we should have the history position of this vertex. However, since the point clouds are inferred from each frame independently, we do not know how each point moves over time since we do not have point correspondence between frames.

To address the problem, we leverage the fact that some objects in the scene are easier to track, and we try to use the motion of these trackable objects to infer motion for the untrackable units. We assume that we know the dense-labeled states of some known fully-actuated shapes like desks and cups connected to the robot arms. Here we will list one specific scenario where a cup of water is poured into another cup. In this case, we have two different types of points: points for fluid and points for cups, we name the states of them in time step $t$ as $V_P^t = \{v_{P,i}^t\}$ and $V_S^t = \{v_{S,i}^t\}$ respectively. For the particle encoder $Q_v$, if the particle belongs to the cups, then the input of particle encoder contains $n_s$ history states before $t_0 : \{V_S^{(t_0-n_s):t_0}\}$. If the particle belongs to the water, then we have no history states, so the input of $Q_v$ is all-zero.

By adding the relative position between receiver and sender points, we can pass the momentum of $V_P$ to $V_S$. Compared with human intuition, we can get an intuitive prediction of the movement of water by simply knowing the past movement of the cup without knowing the past movement of water.

Following Sanchez-Gonzalez et al. (2020), we use the velocity of points and their relative position as inputs to the dynamics module instead of using the absolute positions of the points. This ensures the model is translation-invariant so the learned dynamics model can be shared across different spatial locations.

### B.5 INFERENCE SPEED OF OUR MODEL

The prediction speed of the dynamics module depends on the number of input particles, and it takes around 0.1s for graphs with around 300 nodes in FluidShake and FluidPour, and around 0.2s for scenes with 700+ nodes in GranularPush.

For our visual module, the main time consumption comes from NeRF sampling, it takes 0.2s to sample from a grid space introduced in Section 4 of our paper, this was run in blocks, with block-size=1000, made up 4G of a V100 GPU. And it can be even faster with larger blocks. The sub-sampling process (FPS, segmentation) is fast since they are all written in parallel versions, which takes less than 5ms.

## C POTENTIAL SOCIETY IMPACT

Our work shows the possibility of learning dynamics models from raw sensory inputs, opening up opportunities to automate the design of differentiable physics engines through data-driven learning algorithms. The resulting system can potentially benefit many downstream tasks, including general scene understanding, robotics manipulation, the construction of 3D generative models, and inverse tasks like planning/control and inverse design. Furthermore, predictions from our model are highly interpretable, which makes it straightforward to explain model behaviors and re-purpose the outputs for other downstream applications.

Though data-driven approaches are potentially more scalable with enough data, concerns still exist that it might be hard to ensure the robustness of the model under sensor noise and adversarial attacks. It also becomes less clear how to fully mitigate data biases. Therefore, bringing in advanced techniques from ML robustness will be one critical future avenue to pursue.

