# OpenReview forum: "3D-IntPhys: Learning 3D Visual Intuitive Physics for Fluids, Rigid Bodies, and Granular Materials"
_ICLR.cc/2023/Conference — Submitted to ICLR 2023_

### Official Review · Reviewer_TwJK · 2022-10-24

**Confidence:** 3
**Correctness:** 3
**Technical Novelty And Significance:** 3
**Empirical Novelty And Significance:** 3
**Recommendation:** 6

**Clarity, Quality, Novelty And Reproducibility:**

The paper is very well written and pleasant to read. It was the best paper on my stack - thank you.

I agree with the other reviewers that the authors should be upfront with the limitations of their work.

**Strength And Weaknesses:**

This paper presents a consistent pipeline how to infer future states of physical systems given images only. The pipeline to rely on a NeRF model and uplift instance segmentations into 3D space is a neat idea.

How to infer dynamics from raw pixels is a problem not addressed by prior works.

I have 2 questions remaining: In Sec 3.2, how do you obtain estimates for the velocities since there does not exist explicit correspondences between 3D points across frames? Secondly, what are the point attributes a_i^v? Are these the class labels from the instance segmentation?

**Summary Of The Paper:**

This paper is about dynamics prediction. Given multiple video streams taken from different view points of the same scene, for instance cubes floating in a liquid, the task is to predict future states of the scene. This is a very interesting problem. The proposed solution has a bit of an engineering approach but is reasonable. I generally enjoyed reading this paper and have only a few questions or suggestions how to improve the paper.

**Summary Of The Review:**

The proposed pipeline is reasonable and the problem addressed by the authors is very important and interesting. I am not very familiar with the topic itself, though, so my confidence is not high.

---

> ### Author Response · Authors · 2022-11-19
> **Thank you for your careful reading!**
>
> Thank you for your recognition of our efforts toward learning complex fluid dynamics from visual inputs.
>
> > how do you obtain estimates for the velocities since there does not exist explicit correspondences between 3D points across frames
>
> Here we make an intuitive assumption in our dynamics learning that the momentum is passed from the container to the fluids. And our experiments demonstrated that only using the movement of the container / pusher can also help us learn reasonable dynamics of the fluid particles, their interaction with rigid cubes, and the deformation of the granular materials.
>
> > what are the point attributes a_i^v?
>
> Attribute a_i^v indicates the property of the particle (fluid, rigid, granular).

---

### Official Review · Reviewer_1Jtf · 2022-10-24

**Confidence:** 3
**Correctness:** 3
**Technical Novelty And Significance:** 1
**Empirical Novelty And Significance:** 2
**Recommendation:** 3

**Clarity, Quality, Novelty And Reproducibility:**

The paper is clear and interesting. I think it could be reproduced. However, I question the novelty of the work (see Weaknesses)

**Strength And Weaknesses:**

Strengths:
- The paper provides a possible direction for learning visual dynamics by advocating explicitly utilizing the geometry information.
- The experiments are clear. The proposed method outperforms existing methods.

Weaknesses:
- The proposed two modules are simple combination of existing works. The Perception Module uses the existing PixelNeRF to learn underlying geometry from sparse views. The dynamics simulator applies standard graph-based backbone. There are no much new in the network design.

- The proposed methods use the color information to obtain the masks, which is a very strong assumption and not reasonable. Although the author claims it could also be solved with object segmentation, I think the problem of segmenting fluid, rigid bodies and granular materials is non-trivial (such as labeling, generalization etc) while the proposed method has a very strong reliance on it. This is even more severe when the author claims the methods can achieve "strong generalization". It would be better if the author can show the robustness of the proposed methods with different learned segmentation results. Otherwise I don't think the paper has enough contribution to the problem of learning visual dynamics. It's more like combining two existing methods with a strong and unreasonable assumption to solve a problem.



**Summary Of The Paper:**

The paper presents a method for learning visual intuitive physics models solely from unlabeled multi-view images. Specifically, the method first leverages a conditional NeRF to learn the 3D geometry for the scene, which are then segmented and clustered to produce the 3D point cloud representation for the target fluid, rigid bodies and granular materials. A graph neural network is then applied to learn the dynamics of 3D points. The experiments demonstrate that the methods can produce good visual dynamics and generalize to simple unseen scenes.

**Summary Of The Review:**

I think the assumption of known segmentation is too strong in order to combine the two existing modules. It's unknown how the methods would perform without such an unreasonable assumption. I think more experiments (learned segmentation) and more analysis (robustness to worse segmentation) should be added. I think the paper is below the bar of ICLR.

---

> ### Author Response · Authors · 2022-11-19
> **Thank you for your careful reading!**
>
> Thank you for your careful reading, in-depth feedback, and thoughtful suggestion. The following are responses to your specific concerns.
>
> > The proposed two modules are simple combination of existing works.
>
> We agree with the reviewers that the work is highly relevant to recent work in neural rendering and graph-based dynamics. Yet, we want to clarify that the proposed work aims to tackle the challenging problems of learning visual dynamics from raw images, which neither pixel-NeRF nor graph-based dynamics models alone can solve.
>
> Simply combining the two methods, unfortunately, does not provide a valid solution to the problem since existing point-based dynamics models need to learn from strong supervision provided by 3D ground truth point trajectories, which are hard to obtain in most real setups. For example, in our water experiments, it is impossible for any existing tracking method to successfully track each water particle. To tackle the problem, we propose several new techniques to facilitate dynamics learning without dense correspondence, including momentum passing from containers to fluids and new training loss (e.g., Chamfer distance loss and spacing loss). They allow more robust learning of dynamics models on raw point clouds sampled from the learned occupancy field (instead of the original simulator).
>
>
> > The proposed methods use the color information to obtain the masks, which is a very strong assumption and not reasonable.
>
> We agree with the reviewers that our current method relies on the assumption. Yet, we want to emphasize that the work focuses more on learning complex visual dynamics from images, as opposed to solving object segmentation in general. Learning fluid dynamics from videos is challenging, and only a few existing works exist. NeRF-dy is the closest to us, yet the model's generalization ability is limited. We have shown in the proposed work that we can significantly improve the generalization ability by operating with a hybrid of implicit and explicit, as opposed to pure implicit, 3D representations. We thank the reviewers for pointing out the constraint.

---

> > ### Comment · Reviewer_1Jtf · 2022-11-21
> > **Response**
> >
> > Thank you for your clarification. After reading the author's reply and other reviews, I agree that the idea of generating 3D supervision signals from the video instead of directly using 3D ground truth is interesting. However, I still think the strong assumption makes this method impractical. As said by the author, "Simply combining the two methods, unfortunately, does not provide a valid solution to the problem", but I think the proposed method does not provide a valid solution either, since it relies on many unreasonable assumptions, including assuming the colors of water and other assumptions pointed out by reviewer E7oV. Therefore, I keep my scores.

---

### Official Review · Reviewer_z9jL · 2022-10-25

**Confidence:** 4
**Correctness:** 3
**Technical Novelty And Significance:** 3
**Empirical Novelty And Significance:** 2
**Recommendation:** 5

**Clarity, Quality, Novelty And Reproducibility:**

I think the paper is well-written. Novelty may not be its advantage but there is still enough contribution to the community. The code is provided so I’m not concerned about its reproducibility.

**Details Of Ethics Concerns:**

Not Applicable.

**Strength And Weaknesses:**

Strength:
1) The idea of combining NeRF and graph-based dynamics prediction is interesting. Results are also visually good.
2) The writing is pretty clear and complete. It’s easy to read and understand the main message of the paper. The figures are also nice and well-designed.
3) Code is provided. This could be a good asset to the community.

Weakness:
1) Most of components are adopted from previous works, including image-conditioned nerf and training dynamics prediction using chamfer distance. Getting instance point cloud from segmentation is new but this is relatively easy (color segmentation) and this is not as “imperfect” as the authors claimed in the abstract given the relatively simple scene. It remains unclear if this pipeline can be used in other complex scenarios or even in the real-world.
2) The evaluation is purely based on visual quality while previous work (Li et al. 2021b) also evaluate on control accuracy.
3) What is the merge loss in Figure 2? Does that mean the spacing loss?

**Summary Of The Paper:**

This paper proposes to learn a visual dynamics prediction model in the 3D space. The 3D point cloud representation is learned only from a few images from multiple views (with known camera pose). After that, they use instance segmentation to parse individual objects from the point cloud. Then it trains a graph-based prediction model on the 3D representations, without relying on the ground-truth correspondence, using the chamfer and spacing loss. The proposed method is evaluated using the prediction loss on a few simulation tasks including Pour, Shake, and Push.

**Summary Of The Review:**

In summary, I think some aspects of this paper is worth reading to the community. There are still some discussion and experiments missing as I pointed out in the weakness section and it may still need some work. I’m also happy to discuss and increase my score if the above concerns are addressed.

---

> ### Author Response · Authors · 2022-11-19
> **Thank you for your careful reading!**
>
> Thank you for your careful reading, in-depth feedback, and thoughtful suggestion. The following are responses to your specific concerns.
>
> > Getting instance point cloud from segmentation is new but this is relatively easy (color segmentation) and this is not as “imperfect” as the authors claimed in the abstract given the relatively simple scene.
>
> The reconstructed occupancy measure is not perfect. And color segmentation cannot give us perfect masks, especially when parts of the floating cubes are submerged in the fluid.
>
> > What is the merge loss in Figure 2? Does that mean the spacing loss?
>
> Yes, it means spacing loss. We will revise the manuscript in revision.

---

> > ### Comment · Reviewer_z9jL · 2022-11-21
> > **Response**
> >
> > I thank the authors for their response to my review. As the authors claim, the advantage of their method is that it does not need 3D ground-truth point trajectories. It will be greatly supported and impressive if it can be used in the real-world videos. I understand it’s challenging but I still expect some evidences to show such a transfer is possible. I don’t think the authors clearly answer my question about the how practical this method is. Therefore, I decide to keep my original rating.

---

### Official Review · Reviewer_E7oV · 2022-10-26

**Confidence:** 5
**Correctness:** 2
**Technical Novelty And Significance:** 3
**Empirical Novelty And Significance:** 2
**Recommendation:** 3

**Clarity, Quality, Novelty And Reproducibility:**

The idea of sampling the particle representation from the NeRF model is new. The paper is well-motivated, because being able to learn the simulation from the video data without knowing the ground-truth meshes/particles high desirable. However, the paper does not deliver on this motivation.

The paper does not emphasise that the approach requires a lot of knowledge about the 3D simulation state. The paper also does not clearly mention that actually still requires the ground-truth particle simulations and compute the loss, and therefore the model cannot learn purely from image data. The simulation has to be specifically constructed, such that we can distinguish fluid from other objects (fluid is colored and objects that have to be of different colors)

Additionally, the paper has severe limitations in terms of the systems that it can represent. As stated in section B.4, the fluid particles do not have the history of states, and their velocity is set to zero, and thus the model relies on the *velocities of the objects* to infer the dynamics of the fluid, leading to the incorrect fluid dynamics, as mentioned above. It is unacceptable to put this detail into the last page of the supplementary. The authors should be honest about the applicability and limitations of their model and adequately assess whether the predicted dynamics is, in fact, correct.

Additional questions:

1. Figure 5 shows the comparison of the Chamfer distance between Nerf-DY and the proposed method. How was the Chamfer (point-cloud) distance computed for Nerf-DY, if Nerf-DY performs the dynamics in the latent space and does not have the notion of particles?
2. Table 1 and Figure 4 provide the comparison between the perception modules. Does “Our model” provide only the comparison to PixelNerf? Or are there other parts of the model besides PixelNerf that are included into the “our model” row?
3. Is the particle simulator pre-trained or trained together with the model.
4. Is the proposed model trained only on 1-step prediction or multi-step prediction?

**Strength And Weaknesses:**


## Strengths

This paper is one of the few approaches that are able to perform the dynamics from the video inputs. Unlike the previous approach Nerf-DY, this paper uses a more structured representation of the simulation. That allow the model to generalise to new sizes of containers and different amount of fluid, which is consistent with the findings in the previous papers. The paper connects the two approaches that are known to work well from the previous works: PixelNerf (for a high-res 3D model) and particle-based simulators (for fluid modelling and generalisation). The past works have already shown the strong generalisation to new scenes due to locality of the particle interactions, so it is not surprising to see the similar results in this paper too.

## Weaknesses

There are a few weaknesses in the method that limit the applicability of the work to different systems.

The method requires a major knowledge about the underlying simulation states, which defeats the purpose of learning from video. Specifically:
- Knowledge of the 3D meshes of the robot/container/pusher and their positions/velocities at every time step. This is a strong assumption, because these meshes constitute a large part of the state. The meshes are also notoriously hard to extract from the video in the real-world scenarios.
- Knowledge of fluid density to set the minimum distance for the spacing loss.
- The model relies on the fact that the fluid is coloured and is easy to distinguish from other objects only based on the blue color. Similarly, the objects need to be of different colors, always visible and are easy to track across time.

Another major drawback is that the approach still requires to have the particle simulation as the ground-truth. The mesh/particles and the initial state are notoriously hard to obtain, preventing the previous simulators to train on the real-world data. Therefore, learning directly from videos is desirable. However, the proposed model unable to learn entirely from the video data. Moreover, it requires a corresponding ground-truth particle simulation for every step. It defeats the whole point of learning simulation from videos.

### Representing the correct fluid dynamics.

*Supplementary Section B.4*: “If the particle belongs to the water, then we have no history states, so the input of Qv is all-zero. …. Compared with human intuition, we can get an intuitive prediction of the movement of water by simply knowing the past movement of the cup without knowing the past movement of water”

I don’t think this is true. Consider the case when the we start moving the cup back and forth, and we start recording the simulation when the cup is already moving. The fluid particles already have the prior velocity that is not the same as the velocity of the cup. It is not possible to infer the correct fluid dynamics without the history of fluid particles or velocities.

Another example from the paper is the fluid falling from one container to another.  Velocities of the falling particles depend on gravity and the previous particle velocity. Here we might be able to infer the fluid velocities **only** if we assume that fluid velocities is zero in the beginning of the simulation and we have access to the **entire** history of states. In all other cases, it is not possible to represent the correct dynamics of the fluid.

The fact that the videos look pretty good without the velocity information may indicate that the model exploits some biases from the data that allow to roughly imitate the dynamics under the point-cloud Chamfer loss. Specifically, I suspect that the model misuses the heuristic that the fluid particles have roughly the same as the velocities as the pusher/constrainer. It is visible in the attached videos of the ablations with and without the spacing loss: the particles that do not interact with the pusher are moving with the same speed as the pusher itself. The spacing loss also introduces a strong inductive bias that the fluid density remains constant.

It would be helpful to add the comparison of the dynamics component to the ground truth, starting from the known particle state and evaluated under the Chamfer distance, to compare the correctness of the predicted fluid dynamics. This should be possible to do, as the paper assumes having the ground truth particle simulation.

Authors should be honest about the very limited range of applicability of their model: the fluid velocity can be inferred from the velocities of the surrounding objects, which is a rare case. Even a simple case of the falling block of fluid cannot be represented by the current model.

**Summary Of The Paper:**

The paper proposes a pipeline for predicting the future steps of the physical simulation starting from the video input from multiple cameras. The model builds a Nerf model from several input frames, samples the particles from Nerf and runs the particle-based simulator using the extracted particle representation.


**Summary Of The Review:**

This paper is one of the few approaches that performs the simulation from the video inputs, which is novel.However, this approach is severely limited in the environments that it can model (namely, simulated environments with colored fluids and differently-coloured objects), assumes to know that full mesh and state of the robot/container and relies on the velocities of the known objects to infer the velocities of the fluid. The authors need to be clear about the limitations of their work and the environments where the method is applicable: because fluid particles do not have velocities or past history, it would not be able to produce the correct fluid dynamics. I suggest rejecting the paper due to severe limitations of the approach that are not clearly stated.

---

> ### Author Response · Authors · 2022-11-19
> **Thank you for your constructive comments!**
>
> Thank you for your careful reading, in-depth feedback, and thoughtful suggestion. The following are responses to your specific concerns.
>
> > ‘Another major drawback is that the approach still requires to have the particle simulation as the ground-truth.’
>
> There might be a misunderstanding, as we do not use any particle labels in our pipeline; the only inputs are images. The intermediate particle representation is generated via sampling from the learned occupancy measure using our visual head, which does not need access to the state information from the simulator.
>
> > Learning directly from videos
>
> Since this paper focuses more on learning fluid dynamics, we emphasize learning directly from videos because we do not need dense labels for particles that are widely used by other works. With the help of conditional NeRF, we can better learn 3D representations of a scene purely from multi-view visual observations.
>
> > Known shape information (Robot Arms, Container, Pusher)
>
> Yes, in our settings, we assume that we can get the shape information of containers and pushers, which serve as ‘input signals’ in the dynamic system, as they are easily accessible also in the real world by reading the state information from the robot. Instead, we emphasized learning fluid dynamics, which is more complex, especially purely from visual inputs, without particle-level correspondence labels.
>
> > Known fluid density
>
> Fluid density is a tunable parameter selected based on which levels of granularity we want to model the environment. We selected the parameter based on human intuitions such that it can well capture the general shape of the fluids.
>
>
> > Zero velocity for fluid particles
>
> This is a good point, and we appreciate the example indicated by the reviewer. The assumption is that we can infer the water movement from the container’s movement. We also assume that the initial velocity of water is nearly zero, so the momentum can be gradually passed from the container to the water.
>
> We propose this assumption so that the intuitive physics model can be learned from (1) particles sampled from the neural radiance field, which is not stable (2) point clouds without one-to-one correspondence. The results show that we can learn reasonable dynamics (water poured out from a cup, water falling in the container, cubes moving in water, and granular materials pushed away by a pusher).
>
> > ‘Figure 5 shows the comparison of the Chamfer distance between Nerf-DY and the proposed method. How was the Chamfer (point-cloud) distance computed for Nerf-DY, if Nerf-DY performs the dynamics in the latent space and does not have the notion of particles?’
>
> The dynamics model of NeRF-DY is defined in a latent vector space, where the decoder can predict the occupancy measure of the underlying 3D environment, from which we can also sample in voxel space to get the color / density of each point using the same sampling strategy as our method.
>
> >Table 1 and Figure 4 provide the comparison between the perception modules. Does “Our model” provide only the comparison to PixelNerf? Or are there other parts of the model besides PixelNerf that are included into the “our model” row?
>
> The visual head of our module is PixelNeRF. Here we show that pixel-conditioned NeRF leads to better results than global conditioning / auto-encoder in previous works.
>
>
> >’ Is the particle simulator pre-trained or trained together with the model.
>
> It is trained from scratch with the particles obtained from the visual head, which do not rely on the underlying particle states from the simulator.
>
> > ‘Is the proposed model trained only on 1-step prediction or multi-step prediction?’
>
> It is trained on multi-step prediction, where the step size equals 2.

---

> > ### Comment · Reviewer_E7oV · 2022-11-21
> > **Response**
> >
> > Thank you for providing the clarifications to my questions and criticism. The fact that the model does not use the ground-truth particle simulation is reassuring and makes the paper more convincing. However, the method still requires strong assumptions such as knowing the shape of the container and initialising the velocity to nearly zero. Extracting the simulation from videos is hard, and it is reasonable to make some assumptions, but I would appreciate if the authors were more upfront about them in the main text. As mentioned in the review, I would also like to see the comparison of the learned dynamics of the particles to the ground-truth under the point-cloud loss.
> > For the paper in its current form, I keep my score the same.

---

### Author Response · Authors · 2022-11-19
**General Responce**

We would like to thank all the reviewers for their thoughtful feedback. We are glad that all reviewers acknowledged our efforts toward tackling the challenging problem of unsupervised visual dynamics learning.


> Our technical contribution compared to pixel-NeRF and graph-based dynamics models.

We agree with the reviewers that the work is highly relevant to recent work in neural rendering and graph-based dynamics. Yet, we want to clarify that the proposed work aims to tackle the challenging problems of learning visual dynamics from raw images, which neither pixel-NeRF nor graph-based dynamics models alone can solve.

Simply combining the two methods, unfortunately, does not provide a valid solution to the problem since existing point-based dynamics models need to learn from strong supervision provided by 3D ground truth point trajectories, which are hard to obtain in most real setups. For example, in our water experiments, it is impossible for any existing tracking method to successfully track each water particle. To tackle the problem, we propose several new techniques to facilitate dynamics learning without dense correspondence, including momentum passing from containers to fluids and new training loss (e.g., Chamfer distance loss and spacing loss). They allow more robust learning of dynamics models on raw point clouds sampled from the learned occupancy field (instead of the original simulator).

> Using color-based segmentation.

We agree with the reviewers that our current method relies on the assumption. Yet, we want to emphasize that the work focuses more on learning complex visual dynamics from images, as opposed to solving object segmentation in general. Learning fluid dynamics from videos is challenging, and only a few existing works exist. NeRF-dy is the closest to us, yet the model's generalization ability is limited. We have shown in the proposed work that we can significantly improve the generalization ability by operating with a hybrid of implicit and explicit, as opposed to pure implicit, 3D representations. We thank the reviewers for pointing out the constraint.

---

### Decision · Program_Chairs · 2023-01-20

**Decision:**

Reject

**Justification For Why Not Higher Score:**

As mentioned above, the paper needs to be reworked to be much clearer about the addressed scope of the contributions, the limited practicality of the method, and the exact assumptions that are built into the method. It would require another round of reviews to assess whether these changes have been satisfactorily incorporated into the paper.

**Justification For Why Not Lower Score:**

N/A

**Metareview: Summary, Strengths And Weaknesses:**

This paper tackles the problem of learning a 3D dynamical physics model from multi-view observations of a scene. It follows a multi-stage approach, involving training a NeRF model, sample particles / 3D points from the NeRF representation, and running a particle-based GNN simulator using the extracted particles. It avoids requiring ground-truth correspondence between particles of different time steps by using a Chamfer loss together with a spacing loss. The method, 3D-IntPhys, is evaluated on simulated 3D environments involving manipulation of fluids, rigid bodies and granular materials.

All reviewers agree that this paper is tackling an important and very difficult problem (learning physics from videos) for which currently no general solutions exist – and that the paper is taking small steps toward that direction. The combination of NeRF-based particle representations and a GNN-based simulator is novel and interesting. The experimental results demonstrate that the proposed method addresses the considered tasks well. The writing is very clear and the paper is generally of high quality.

The main concerns highlighted in the original reviews and the discussion period are that the paper makes very strong and limiting assumptions about the considered environments; some of these assumptions are not clearly discussed upfront, or are completely missing from the main text of the paper. These assumptions include: known velocity of the fluid (experiments start from nearly static fluid), object segmentation (by color), known meshes/state for some aspects of the environment (container, robot, pusher), and knowledge of fluid density for the spacing loss. Overall, concerns were raised that assumptions make the method very impractical for use beyond the demonstrated synthetic settings.

I agree with the reviewers that the paper needs to be reworked to be much clearer about the addressed scope of the contributions, the limited practicality of the method, and the exact assumptions that are built into the method. It would require another round of reviews to assess whether these changes have been satisfactorily incorporated into the paper. With the current set of assumptions, the scope/applicability of the method is rather limited. If the authors could demonstrate some real-world results or results for an environment where the ground-truth state of part of the system is not known, while being upfront about any remaining assumptions, the paper would be a great fit for a conference like ICLR/ICML/NeurIPS. In its current form, my recommendation is to reject the paper.


**Summary Of Ac-Reviewer Meeting:**

During the discussion, every reviewer was asked to state/reiterate reasons for rejecting/accepting the paper, after which we discussed the main aspect for which there was originally no consensus: to which degree can the paper still be reworked by the authors to properly discuss and/or address the limiting assumptions made by their method.

The discussion concluded with everyone agreeing that properly addressing/highlighting the limitations of the method (i.e., the assumptions about the environment made by the model) would require a (major) rework of the paper and another round of reviews, which is a reason to reject the paper.

Detailed meeting notes:

Initial views about paper before discussion:

Reviewer E7oV:
* Recommendation: reject
* Reasons for accept: learning dynamics from videos is really hard – paper takes small steps toward that direction
* Reasons for rejection hide a lot of assumption in the appendix, assume know velocity of fluid – always start from static fluid. Cannot start from moving fluid. Hard to reproduce since most details are hidden in supplementary; Limitations: color segmentation, mesh, velocity

Reviewer TwJK:
* Original recommendation: strong accept
* Limitations are hidden, but does not justify reject; Authors should very clearly state limitations, Paper should be accepted since it addresses a very hard problem. Not supposed to solve the problem fully, but opens the door to address those limitations
* Great basis for future work to address limitations and build on top

Reviewer 1Jtf: recommend rejection
* Cannot get a clear message from the paper; we know that with 3D supervision we can learn dynamics of particles;
* Very strong assumptions so that not a clear message can be gained from the paper. Message: we can use NeRF to get pseudo-ground truth.
* Paper does not contribute much since they cannot solve the problem.
* Assumptions are not practical

Reviewer z9jL:
* Some advantages of the paper: combination of NeRF + GNNs is very interesting; doesn’t require explicit state
* Gave borderline reject: is method practical? Authors claim that advantage of their method is that they learn from the real world -> should show results from real-world videos
* Impression is that they have very clean separated scene like in simulation; no noise on color segmentation etc. — doubts about practicality
* Paper is very closely related to previous work (Li et al. 2021b) – show that MPC using such a prediction model
* Comparison to MPC was asked for but did not respond
* If real world result and/or robotics control (simulation) -> weak accept or accept

Reviewer E7oV:
* Agrees with Reviewer 1Jtf; point cloud loss and chamfer distance isn’t a fully novelty [after AC mentioned that Point cloud matching + chamfer distance seems novel]
* Chamfer loss and spacing loss are a fix to a problem that they don’t have correspondence and velocity; sounds like a hack rather than a novelty;
* Will be really hard to extend to something more general

After discussion:
* Reviewer 1Jtf: Still prefer to reject
* Reviewer E7oV: Still keep reject; don’t know if they would properly address those limitations; would require major rework of the paper; would like to see updated version after that
* Reviewer TwJK: Now shares concern that authors might not properly address those limitation; reason to reject
* Reviewer z9jL: Lean towards rejection; evaluated “as is” and not conditional on edits; borderline reject